# The Effect of General Anesthesia on the Outcome of Root Canal Treatment in Pediatric Patients—A Retrospective Cohort Study

**DOI:** 10.3390/children10030520

**Published:** 2023-03-07

**Authors:** Shlomo Elbahary, Eyal Rosen, Sohad Haj-Yahya, Maysa Ghrayeb Elias, Shany Talmi, Igor Tsesis, Hagay Slutzky

**Affiliations:** 1Department of Endodontics, Maurice and Gabriela Goldschleger School of Dental Medicine, Tel Aviv University, P.O. Box 39040, Tel Aviv 6997801, Israel; 2Department of Prosthodontics, Goldschleger School of Dental Medicine, Tel Aviv University, P.O. Box 39040, Tel Aviv 6997801, Israel

**Keywords:** primary root canal treatment, outcome, pediatric, general anesthesia, immature tooth, pulp therapy

## Abstract

This study aimed to evaluate the effect of general anesthesia (GA) on the 1-year outcome of Root Canal Treatment (RCT) performed in pediatric patients and to compare it to the outcome of RCT in pediatric patients without GA. Patients admitted for RCT in permanent dentition in a public hospital, dated 2015 to 2020, age 8–15 with a minimum of one year follow-up period, were included in the study. The sample consisted of 326 teeth from 269 patients treated by a single operator, with a recall rate of 81%. Overall, 124 teeth were treated under GA and 142 teeth were without GA. The mean follow-up time was 31.5 months. Data underwent statistical analysis and the significance threshold was set for *p* < 0.05. Of the total cases, 90% showed favorable outcomes. A significantly higher favorable outcome was seen in the GA group than in the non-GA group (98% and 85%, respectively, *p* < 0.001). The outcome was significantly affected by the type and quality of the coronal restoration, degree of root development, and lesion size (*p* < 0.05). According to the current study, in uncooperative pediatric patients, a more favorable outcome of root canal treatment can be obtained under GA than LA if the procedure is carried out with immediate restoration.

## 1. Introduction

Effective control of anxiety and pain plays a crucial role in pediatric patients’ compliance and general adherence to pulp therapy and dental treatment [1]. Pain management in pediatric patients includes local anesthesia, topical anesthetic, computer-controlled local anesthetic delivery (CCLAD) systems [2] and cryotherapy to minimize pain and provide a positive dental experience for the patient [3]. General anesthesia (GA) or deep sedation are often required to complete necessary dental radiographs and/or treatment in young, uncooperative children [4]. Despite a low incidence of adverse outcomes, treatment with GA or deep or moderate sedation can be safely and efficiently completed in a dental office setting by well-trained providers following established protocols and guidelines [4,5,6]. Indications for employing general anesthesia for dental treatment include the inability to cooperate with conventional chairside dental treatment; extensive treatment needs, exceeding that which the patient could tolerate; the need for complex procedures; or allergy to local anesthesia [7,8].

Dental treatment under GA has several advantages: It does not require the patient’s cooperation and the patient is unconscious and non-responsive to pain. In addition, all necessary dental procedures can be completed in just one visit while using pharmacological sedation such as Midazolam, which is commonly used as an oral sedation agent in children and has a short-acting time of action, limiting us to only perform short dental procedures [9,10]. On the other hand, there are more postoperative complications than in the local anesthesia (LA) [9]. The most common dental treatments performed for children under GA include restorative treatments, pulp therapy, minor surgeries, and extractions. Khodadadi et al. [11] published a study from 2018 designed to assess the failure rate of pediatric dental treatment on primary teeth under general anesthesia. It concluded that the overall failure rate of dental treatments performed on pediatric patients under GA was 6.59%, with pulp therapies having the lowest rate of failures.

Treating pulpal infection in young permanent teeth in children and adolescents presents a unique challenge to dental clinicians [12]. Reported success rates for conventional endodontic therapy in adults range from 40% to 97%, and this variation is due to the lack of a clear definition of the success and failure of endodontic treatment in these studies [13]. This retrospective observational cohort study was designed to compare the outcome of RCT under GA or LA in the permanent teeth of children with mature and immature teeth. Our null hypothesis was that no difference would be found between the groups.

## 2. Materials and Methods

This retrospective observational cohort study of RCT outcome in association with GA in patients with behavioral problems and inability to cooperate was conducted after approval from the institutional ethics committee board at Clalit Medical Services (approval number: 0065-21-COM2) and the Tel-Aviv University ethics committee board (approval number: 0002711-3). The paper was written in accordance with “Strengthening the reporting of observational studies (STROBE)” recommendations [14].

### 2.1. Source of Data

All patient data was taken from the “Clalit Smile” database, which holds electronic medical records.

### 2.2. Study Sample

Patients were aged 8–15 years, admitted for RCT in permanent dentition between 2015 and 2020, with a minimum of one-year follow-up. All patients received RCT by a single operator (S.E), who is a board-registered specialist in endodontics. Primary teeth, cracked teeth, teeth exhibiting signs of resorption, teeth with dental anomalies, teeth with poor prognosis [15], and teeth with root perforations were excluded.

### 2.3. Cohorts

The GA cohort group consisted of pediatric patients who received RCT under GA. The patients who received RCT but without GA were considered the non-GA cohort group.

### 2.4. Preoperative Diagnosis and Outcome Assessment

All patients referred for treatment were classified as Frankl 1–3 [16]. All patients had at least two attempts at treatment under LA with the aid of nitrous oxide. Patients who did not cooperate after two attempts and were classified as Frankl 1–2 [16] were referred for treatment under GA. The diagnosis determined before the RCTs was based on the patient’s complaints, the clinical signs and symptoms, and the Rx evaluation (assessed by PA radiography taken with a film holder and positioning arm). These criteria [17] (“AAE Consensus Conference Recommended Diagnostic Terminology”, 2009) were also used at the follow-up session.

### 2.5. Root Canal Treatment Method

A single operator (SE) used an operative microscope (SOM32, Karl Kaps GmbH & Co., KG, Wetzlar, Germany) for all treatments, which were performed using the most recent crown-down techniques under a rubber dam. A proper cleansing of the pulp chamber and the coronal part of each tooth to be treated was performed, and the cavity preparations were made with a no. 330 high-speed bur (Komet, Lemgo, Germany) under water coolant. Following completion of the cavity outline and the access to the pulp chamber, no. 5 carbide round bur (Komet, Lemgo, Germany) in a slow-speed handpiece was used to complete the preparation of the pulp chamber and expose the canal orifices. The coronal part of the root canals was then prepared using rotating instruments (Gates Glidden Drills, Kerr, Brea, California, United States. and Protaper system, Dentsply Sirona, Tulsa, Oklahoma, United States.) and ultrasonic tips (EMS, Nyon, Switzerland). The root-canal system was prepared with a stainless-steel K-file and NiTi engine-driven (X-smart, DENTSPLY Sirona, Tulsa, OK, USA) instruments (Protaper gold/next, DENTSPLY Sirona, Tulsa, OK, USA) and copious irrigation with 5% sodium hypochlorite (Prime Dental, Bhiwandi, India) and EDTA (Clarben, Madrid, Spain). The root canal length was determined by an electronic apex locator (RootZX 2, Morita, Tokyo, Japan). The sequence of instrumentation was strictly dependent on the root-canal morphology. All teeth were prepared and sealed in one visit and were all sealed with gutta-percha (IM3, Washington, DC, USA) using lateral compaction with an “AH plus” sealer (DENTSPLY Sirona, Tulsa, OK, USA). The sealing limit was set to within 0.5 to 1 mm from the radiographic apex, with a slight extension beyond this sealer limit tolerated. At the end of the treatment, a complete adhesive reconstruction (direct), with intracanal retention, if needed, was made to avoid coronal leakage.

### 2.6. Outcome Assessment

Radiograph Execution and Examination: All follow-up appointments included in this study used customized film holders (Kerr, Brea, CA, USA) and X-ray film (Prime Dental, Bhiwandi, India) to expose radiographs using the paralleling technique. The follow-up examination included two radiographs and a clinical examination. Radiographs were also taken before and at the end of treatment.

All collected radiographs were independently examined under magnification twice (at an interval of almost 20 days) by two calibrated endodontics independent observers. In situations where the two observers disagreed, the lowest score was recorded. Scoring was based on the periapical index (PAI) [18], an established system for assessing apical periodontitis in radiographs. The outcome was classified as favorable if there were no radiographic signs of apical periodontitis (PAI score < 3) and no clinical signs and symptoms. Any other result was classified as “unfavorable”. The whole tooth was considered the unit of evaluation. The coronal restoration was also examined as follows: Good restoration consisting of any permanent restoration that appeared intact radiographically. Poor restoration consisting of any permanent restoration with radiographic signs of overhangs; recurrent decay or open margins [19].

### 2.7. Statistical Analysis

G*Power 3.1.9.4 software was used to estimate two tails sample size using *t*-test—linear bivariate regression for two groups with the following parameters: Δintercept = 2, α err prob = 0.05, power (1-β err prob) = 0.95; the no centrality parameter δ = 3.6348353 with critical *t* = 1.9701536. For actual power of 0.9515341, each of the two groups was to include at least 119 individuals. Statistical analysis was made using SPSS (version 22; SPSS Inc, Chicago, IL, USA). The analysis included overall 2 chi-square tests. The first chi-square was performed to check the correlation between the two assessors. The second test was also a chi-square test of proportions and Fisher’s exact test and provided an analysis of the outcomes. The tests were performed as one-tailed and interpreted at the 5% significance level, and the dependent variable was the dichotomous outcome, favorable versus unfavorable. The null hypothesis (H0) was that the outcome and success rate of RCT under DGA will not differ from the outcome of RCT under LA.

## 3. Results

The sample consisted of 326 teeth from 269 patients, with a recall rate of 81% (218 patients), where 124 teeth were treated under GA and 142 teeth under LA. The follow-up time ranged from 12 to 91 months, with a mean of 31.5 months and a standard deviation of 21.5 months.

The teeth were characterized according to preoperative variables in Table 1.

Young patients and females were significantly more likely to be treated under GA (*p* < 0.001).

The total favorable outcome was 90% and 10% unfavorable. The favorable outcome in the GA group was significantly higher than in the non-GA group (97.5% and 84.5% healed, respectively, *p* < 0.001).

The total outcome was significantly affected by the type of restoration and quality (*p* < 0.05); teeth with a stainless-steel crown showed a 94.3% favorable outcome and 90.1% favorable outcome for teeth with composite restoration (*p* < 0.01). A 93.9% favorable outcome was shown in teeth with good restoration, while a favorable outcome was 50% for teeth with failed restoration (*p* < 0.04).

There was a significantly higher favorable outcome for fully formed teeth than immature ones (93.9% vs. 83.9%, respectively, *p* = 0.004). Lesion size also significantly affected the outcome, with a favorable outcome of 80.6% for teeth with a lesion larger than 5 mm; 90.6% for teeth with a lesion smaller than 5 mm; and 94.9% for teeth with no periapical lesion (*p* < 0.05).

In contrast, there was no significant association between favorable outcomes and age, tooth type, pulp diagnosis, sinus tract, root resorption, or follow-up period. The associations between all parameters and outcome status are presented in Table 2.

## 4. Discussion

Molar teeth have been associated with a lower chance of survival and with the greatest risk of developing post-treatment disease; this may be attributable to the complex anatomy of molars and their accessibility and technical difficulties in treating them, as well as to the heavy occlusal stresses acting on these teeth [20]. Concerning pediatric endodontics, the prevalence of young permanent posterior teeth with pulp involvement was found to be very high (36.9%) [21]. A study from 2018 [11] found that endodontic treatment performed on permanent teeth of 6–18-year-olds showed a high survival and low failure rate during a follow-up period ranging from 1 month to 8 years. Notably, endodontic procedures can be complex and challenging to undertake under general anesthesia (GA) [22]. This may be due to limited mouth opening, tongue protrusion due to intubation, the short duration of the session, uncertain diagnosis due to the inability of patients to describe their symptoms, and the need to complete a maximum amount of treatment at each session. Some argue that the conditions for endodontics under GA are so difficult that the operator may frequently have to choose between extraction or a series of technical compromises [6]. Alsaleh et al. (2012) [6], who evaluated the quality of endodontic treatment under GA (255 teeth in patients with special needs) and compared it to root canal treatments (RCTs) under local anesthesia (264 teeth), reported that the results were comparable and satisfactory in 63% of the cases. However, the quality was lower for molars in both groups. Similarly, a study by Cousson et al. (2014) [22] in the special needs unit of the University Dental Hospital of Clermont-Ferrand, revealed that 87% of 225 permanent teeth could be considered to have a successful outcome of endodontic treatment under GA after a follow-up period of 1 month to 2 years. In 2017, Chang et al. [23] also reported a favorable 5-year survival rate of 89.9% after a single visit to endodontic and restoration treatment under GA for 381 teeth in 203 special needs patients. These results were supported by Chung et al. [24] (2019), who evaluated the outcome of RCTs under GA in 271 special needs patients with a mean age of 26 years, with a follow-up period of more than 12 months, and observed complete periapical healing without clinical signs in 81.5% of the teeth. A recent observational retrospective cohort study that compared the survival rate between RCTs under GA (280 teeth) or LA (217 teeth) for patients with special needs reported a cumulative survival rate of 87.68% and 74.51%, respectively, in the general anesthesia group and non-general anesthesia group over the nine-year follow-up period [25].

Although previous studies have reported that endodontic treatment may be performed under GA, and also on the information about the outcome of dental treatment and particularly in relation to endodontics [9,24].

To our knowledge, there is no previous study in the literature that compares the outcome of primary RCT performed under GA and non-GA treatment. This study’s results demonstrate a significantly higher rate of favorable outcome (97.5%) for RCT under GA as opposed to the non-GA group (84.5%), with an average of 31.5 months follow-up. This is a higher outcome rate than reported by Cousson et al. [22] for primary RCT under GA, where 85% of outcomes were evaluated as successful, while 10% were uncertain, and 5% failed after a follow-up of 24 months. The null hypothesis was that no difference would be found between the groups, and it was rejected. Chung et al. [24] similarly reported an 81.5% success rate using the same criteria. The current study’s 90% favorable outcome rate is in accordance with the 87% rate found in a Toronto study [26] with equivalent outcome criteria. The overall drop-out rate in the current study was low (19%). The most common reasons for missing follow-up appointments were that patients refused to arrive or could not be contacted. A less common reason was the need for more retrievable radiographs from the files.

Interestingly, a study by Ricucci [27] in 2011 with a single operator (as in this current study) reported a similar success rate (88.6%) to that seen in this study. These results are also in accordance with the success rates of RCT performed by general dentists, which are recorded as between 31% and 96% [8]. Therefore, the fact that a single endodontic specialist who is highly skilled in endodontic treatments under GA performed all the RCTs in this study, may explain the study’s high success rate. Endodontic treatment by a specialist is more successful [28], despite the compromises that they may be forced to make by the GA-associated difficulties already described [6] (restricted mouth opening, tongue protrusion, presence of intubation tube, and time constraints). Several measures concerning instrumental procedures, determination of apical length, and coronal restoration may be modified during RCT under GA to minimize treatment time. Notably, all the RCT treatments in this current study were carried out using rotating Ni-Ti instrumentation [29] and an apex locator [30,31], which helped us to reduce the time under GA and to improve the safety and efficacy of the treatment. Following previous studies [19,32], the quality of the coronal restoration had a significant effect on the outcome, with a more favorable outcome for teeth with a good restoration. In this study, almost all the molar teeth were restored using core and SS crown immediately after the root canal sealing, which may have contributed to RCT’s highly favorable outcome rate. This procedure [33,34,35] has been reported to yield a high success rate when performed in children under GA.

Conventional endodontic treatment in young permanent dentition has long-term effects since there is no further root formation. This leaves the root with short, thin dentin walls, which increases the risk of cervical fractures and endangers the survival of the entire tooth [36,37,38]. Furthermore, the open apices of immature teeth make it harder to perform good RCT without complications, such as the extrusion of root-filling material into the periapical tissue [39]. A further complication is that while the application of MTA in the root canal is known to facilitate an apical seal, it does not promote continued root development. Accordingly, most MTA studies do not report significant continued root formation and maturation [39]. These issues may explain this study’s higher favorable outcome rate with fully formed teeth compared to immature teeth (93.9% vs. 83.9%, respectively).

The current study considered only primary RCTs. Although the pulp was evaluated in each patient, the reliability of the tests may have been compromised due to a lack of cooperation on the part of the child. Moreover, pulp maturation is known to continue for up to 3 years after the eruption, during which time, the sensitivity of vital pulp testing is imprecise [40,41]. While fully appreciating that endodontic diagnosis is mandatory prior to RCT [42,43], the severe behavior that required the use of GA in some children also precluded an accurate preliminary diagnosis, even though it was obvious that endodontic treatment was essential and must be carried out to prevent extraction or disease [44,45]. In the present study, there was a statistically significant difference in the outcome observed in large (>5 mm), small (<5 mm), and no lesions (80.6%, 90.6%, and 94.9%, favorable outcome, respectively). Previous studies have described a positive relationship between the size of PA lesions and bacterial density and diversity inside the root canal [46,47], which could impact the treatment outcome. Teeth with small PA lesions have been demonstrated to yield higher favorable outcome rates in some studies [46,47,48], although others found no statistically significant differences between small and large PA lesions [15,26,49].

In the end, the practice of endodontic treatment under GA should be encouraged when indicated, and avoiding GA, when required, may have a negative impact on pediatric patients’ oral health and quality of life [50,51]. Since prosthetic and implant rehabilitation may not be suitable for pediatric patients [8,35], the practitioner is left with the two main treatment options being endodontic treatment or extraction. The first, with the potential of retaining functional teeth, is preferable despite the possible challenges of performing high-quality endodontic treatment in young and maybe uncooperative patients. Another issue the article did not deal with in, but it is just as important, is the need for more awareness among parents about dental or root canal treatment under GA. Parents sometimes need to understand the consequences and risks involved in performing treatment under GA and the advantages, disadvantages, and outcome [52].

## 5. Conclusions

According to the current study, in uncooperative pediatric patients, a more favorable outcome of root canal treatment can be obtained under GA than LA, if the procedure is carried out with immediate restoration.

## Figures and Tables

**Table 1 children-10-00520-t001:** Associations between preoperative parameters and GA.

Characteristics	GA	No GA	*p*-Value
*n*	124 (46.6)	142 (53.4)	
Gender, *n* (%)			<0.001
Male	29 (24.0)	78 (55.3)	
Female	92 (76.0) *	63 (44.7)	
Age, Mean ± SD	12.0 ± 1.8 *	13.6 ± 4.2	<0.001
Lesion size			0.029
<5 mm	13 (41.9)	40 (34.5)	
≥5 mm	2 (6.5)	34 (29.3)	
none	16 (51.6)	42 (36.2)	

* Statistical significance.

**Table 2 children-10-00520-t002:** Associations between parameters and outcome status.

Characteristics	Favorable	Unfavorable	*p*-Value
GA, *n* (%)			<0.001
*Yes*	117 (97.5)	3 (2.5)	
*No*	120 (84.5)	22 (15.5)	
Gender, *n* (%)			0.012
*Male*	90 (84.9)	16 (15.1)	
*Female*	146 (94.2)	9 (5.8)	
Age, Mean ± SD	12.9 ± 3.4	12.3 ± 3.2	0.358
Tooth type, *n* (%)			0.789
*Premolar*	20 (90.9)	2 (9.1)	
*Molar*	97 (91.5)	9 (8.5)	
*Incisor*	120 (88.9)	15 (11.1)	
Lesion size			0.048
*<5 mm*	48 (90.6)	5 (9.4)	
*≥5 mm*	29 (80.6)	7 (19.4)	
*none*	56 (94.9)	3 (5.1)	
Sinus tract, *n* (%)			0.135
*Yes*	23 (82.1)	5 (17.9)	
*No*	214 (91.1)	21 (8.9)	
Root formation, *n* (%)			0.010
*Full*	154 (93.9)	10 (6.1)	
*Open*	73 (83.9)	14 (16.1)	
Root resorption, *n* (%)			0.098
*Yes*	39 (84.8)	7 (15.2)	
*No*	185 (92.5)	15 (7.5)	
Restoration type, *n* (%)			<0.001
*Acryl Crown*	2 (66.7)	1 (33.3)	
*SS crown*	83 (94.3)	5 (5.7)	
*Composite*	146 (90.1)	16 (9.9)	
*Amalgam*	2 (66.7)	1 (33.3)	
*IRM*	0	2 (100.0)	
*PFM*	4 (80.0)	1 (20.0)	
Restoration quality, *n* (%)			0.004
*Good*	205 (91.1)	20 (8.9)	
*Caries*	29 (90.6)	3 (9.4)	
*Failed*	3 (50.0)	3 (50.0)	
Follow-up period, *n* (%)			0.615
*≤12 months*	41 (91.1)	4 (8.9)	
*13–24 months*	97 (88.2)	13 (11.8)	
*25–36 months*	40 (95.2)	2 (4.8)	
*≥37 months*	58 (89.2)	7 (10.8)	

## Data Availability

The data presented in this study are available on request from the corresponding author. The data are not publicly available due to ethical considerations.

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
