# Peer review of "The Effect of General Anesthesia on the Outcome of Root Canal Treatment in Pediatric Patients—A Retrospective Cohort Study"

_children, 2023, doi:10.3390/children10030520_

Round 1
Reviewer 1 Report
Line 42 “The maximum time available for surgical work when the most common pediatric enteral moderate sedation technique with midazolam can be as short as 20–30 minute” is a copied sentence
Line 121 ProTaper system is not usually used to pulp chamber and coronal part of the tooth.
Line 128 why the sequence of instrumentation was strictly dependent on the root-canal morphology? What’s the meaning?
Line 132 what’s the meaning of sealer limit tolerated?
Line 133 How did they realized a indirect complete reconstruction in one visit treatment protocol?
Line 165 the table 1 only pre-operative variables has been showed
Line 177 What’s the meaning of good restoration, basing on what kind of clinical/radiographical characteristics did they judge a good restoration? Lack of bibliography.
Line 180 in the exclusion criteria they exclude teeth requiring apexification (line 100), but in the results section they included immature teeth why?
Line 193 there no study in the literature that compare the outcome of RCT under GA and non GA?
Author Response
Dear reviewer,
Thank you very much for the constructive comments, which I am sure will make the article more readable and accurate.
I took the comments seriously, and I am attaching an updated version with the necessary changes.
Comment :Line 42 “The maximum time available for surgical work when the most common pediatric enteral moderate sedation technique with midazolam can be as short as 20–30 minute” is a copied sentence
Answer: The sentence has been rephrased
Comment: Line 121 ProTaper system is not usually used to pulp chamber and coronal part of the tooth.
Answer: the issue was addressed in the text in lines 126-130, and a more detailed description was added to the text.
Comment: Line 128 why the sequence of instrumentation was strictly dependent on the root-canal morphology? What’s the meaning?
Answer: Even though we have several Files in the Protaper gold kit, in sosme cases when we have different root morphology between teeth, for example narrower or curved canals, we will choose a specific file to use before the previous one and vice versa, or choose a manual over rotary preparation in some stage. and this is of course according to the manufacturer's instructions.
Comment: Line 132 what’s the meaning of sealer limit tolerated?
Answer: In general, we would like the sealer to remain only within the root canal system, however, there are cases of sealer exiting the root canal, as shown in previous studies, the extrusion of a minimal amount as described here, does not affect the RCT outcome. While a massive outflow of material or overextension of the gutta percha will result in harm to the prognosis and therefore such cases were not included.
Comment: Line 133 How did they realized a indirect complete reconstruction in one visit treatment protocol?
Answer: The issue was addressed and rephrased. Only direct restorations were made.
Comment: Line 165 the table 1 only pre-operative variables has been showed
Answer: The title and line were corrected accordingly.
Comment: Line 177 What’s the meaning of good restoration, basing on what kind of clinical/radiographical characteristics did they judge a good restoration? Lack of bibliography.
Answer: the issue was added and addressed in methodology line 162 according to Tronstad et al.
Comment: Line 180 in the exclusion criteria they exclude teeth requiring apexification (line 100), but in the results section they included immature teeth why?
Answer: the issue was addressed in the methodology. Line 105
Comment: Line 193 there no study in the literature that compare the outcome of RCT under GA and non GA?
Answer: As mentioned, as far as we know, we have not found a study comparing the results of root canal treatment performed by the same therapist under general anesthesia and without general anesthesia.
Reviewer 2 Report
Dear Authors,
I have been invited to review your work entitled “The effect of general anesthesia on the outcome of root canal treatment in pediatric patients – a retrospective cohort study”. The study was well conducted, however there are some issues that deserve revisions for the acceptance of the manuscript. Please, provide a point-by-point response, highlighting the corrections with a different color mark for each reviewer.
Abstract
• The abstract should be structured in the IMRaD format, without headings.
• You should add that data underwent statistical analysis with significance threshold.
Introduction
• The introduction should briefly introduce the topic of the research. Many considerations should be moved to the discussion section (e.g. lines 57-84).
• Before talking about general anesthesia, a short paragraph on different attempts to reduce pain during local anesthesia in pediatric patients could be added(DOI: 10.22514/jocpd.2023.002; DOI: 10.22514/jocpd.2022.018).
• The null hypotheses of the study should be added and then rejected/accepted in the discussion.
Materials and methods
• Line 108: it is Frankl behavioural scale, not Frankel.
• You can avoid repeating city and state of the materials cited starting from the second citation. You can just write the name of the manufacturer.
• Where is the sample size calculation? A numerical calculation should be provided using data from previous studies (primary endpoint, expected mean, expected mean difference,standard deviation, alpha and beta errors).
Results
• You talk about “association” when you describe the results of the study, but you did not specify in the results the statistical test used. Please, specify in the text to which statistical test the P values refer to. Please, write
• Write the correct percentages (from table 2, I desume) in the text of the results (lines 169-187).
Discussion
• Please, write the correct percentages also in this section.
• Another aspect that should be taken into consideration is that one of parents’ concern about general anesthesia, as a correct informed consent should be provided to explain all the possible effects of the procedure. (DOI: 10.22514/jocpd.2022.030).
• Adverse effects of GA were not properly investigated in this research, this is a limitation of the study.
• Other limitations could be added.
General issues
• The references do not follow the template of the Dentistry Journal. Please, follow the
Instruction for authors and properly format the reference list.
• Extensive English spelling issues should be solved.
• Grammar and punctuation errors are widespread among the manuscript. Please, revise it properly.
• Avoid personal writing among all the manuscript.
Thank you for the effort.
Author Response
Dear reviewer,
Thank you very much for the constructive comments, which, I am sure will make the article more readable and accurate.
I took the comments seriously, and I am attaching an updated version with the necessary changes
Abstract
Comment:• The abstract should be structured in the IMRaD format, without headings.
Answer: The issue was addressed.
• You should add that data underwent statistical analysis with significance threshold.
Answer: significance threshold is specified.
Introduction
- The introduction should briefly introduce the topic of the research. Many considerations should be moved to the discussion section (e.g. lines 57-84).
• Before talking about general anesthesia, a short paragraph on different attempts to reduce pain during local anesthesia in pediatric patients could be added(DOI: 10.22514/jocpd.2023.002; DOI: 10.22514/jocpd.2022.018).
• The null hypotheses of the study should be added and then rejected/accepted in the discussion.
Answer: The issue was addressed and null hypothesis was added (line 28,97)
Materials and methods
• Line 108: it is Frankl behavioural scale, not Frankel.
The name was corrected
You can avoid repeating city and state of the materials cited starting from the second citation. You can just write the name of the manufacturer.
• Where is the sample size calculation? A numerical calculation should be provided using data from previous studies (primary endpoint, expected mean, expected mean difference,standard deviation, alpha and beta errors).
The issue has been corrected and added to the statistics section. However full calculations will be uploaded as a statistics file as supplementary material
Results
• You talk about “association” when you describe the results of the study, but you did not specify in the results the statistical test used. Please, specify in the text to which statistical test the P values refer to. Please, write
• Write the correct percentages (from table 2, I desume) in the text of the results (lines 169-187).
Answer: Correct percentage were added.
Discussion
• Please, write the correct percentages also in this section.
• Another aspect that should be taken into consideration is that one of parents’ concern about general anesthesia, as a correct informed consent should be provided to explain all the possible effects of the procedure. (DOI: 10.22514/jocpd.2022.030).
• Adverse effects of GA were not properly investigated in this research, this is a limitation of the study.
• Other limitations could be added.
The correct percentage were added. And the issue was addressed (line 297)
General issues
• The references do not follow the template of the Dentistry Journal. Please, follow the
Instruction for authors and properly format the reference list.
• Extensive English spelling issues should be solved.
• Grammar and punctuation errors are widespread among the manuscript. Please, revise itproperly.
• Avoid personal writing in all the manuscript.
The reference styles were changed using the Endnote MDPI ref Style.
Writing issues were addressed.
Round 2
Reviewer 2 Report
Dear Authors,
Thank you for providing the revised version of your manuscript. I have noticed that the modifications were not completely performed: please, revise the manuscript carefully, otherwise it will be rejected. Here are enlisted the points that still need revision.
· You should write in the abstract that “Data underwent statistical analysis” and “the significance threshold was set for p < 0.05”.
Introduction
- This point was not addressed: The introduction should briefly introduce the topic of the research. Many considerations should be moved to the discussion section (e.g. lines 57-84).
· Please, consider all the suggested references, as CCLADs are widely used in pediatric dentistry to administer local anesthesia (DOI: 10.22514/jocpd.2023.002).
Materials and methods
· Also in line 115 there is “Frankel”, it is “Frankl”. It is implied that you should correct the term all over the manuscript.
· You can avoid repeating city and state of the materials cited starting from the second citation. You can just write the name of the manufacturer. Example: “The root-canal system was prepared with a stainless-steel K-file and NiTi en- gine-driven (X-smart, DENTSPLY Sirona, Tulsa, Oklahoma, United States.) instruments (Protaper gold/next, DENTSPLY Sirona, Tulsa, Oklahoma, United States.).” “The root-canal system was prepared with Protaper gold/next stainless-steel K-file (DENTSPLY Sirona, Tulsa, Oklahoma, United States) and X Smart NiTi engine-driven (DENTSPLY Sirona).
· The sample size calculation was not correctly performed. Still, a numerical calculation should be provided using data from previous studies (primary endpoint, expected mean, expected mean difference, standard deviation, alpha and beta errors).
Results
· You talk about “association” when you describe the results of the study, but you did not specify in the results the statistical test used. Please, specify in the text to which statistical test the P values refer to. Please, write
Discussion
· The null hypothesis was not rejected/accepted in the discussion.
General issues
· The references do not follow the template of the Dentistry Journal. Please, follow the Instruction for authors and properly format the reference list.
· Extensive English spelling issues should be solved.
· Grammar and punctuation errors are widespread among the manuscript. Please, revise it properly.
· Avoid personal writing among all the manuscript. This aspect is important. No personal pronouns please (our work, we did, and so on).
